# Clinical Influence of Bile Duct and Duodenum Preservation on Zinc Absorption and Remnant Pancreatic Volume in Duodenum-Preserving Pancreatic Head Resection for Low-Grade Malignant Pancreatic Tumors

**DOI:** 10.3390/cancers17132217

**Published:** 2025-07-02

**Authors:** Yoshiki Kunimura, Hiroyuki Kato, Satoshi Arakawa, Masahiro Shimura, Takahiro Tashiro, Daisuke Koike, Hidetoshi Nagata, Yuka Kondo, Hironobu Yasuoka, Takahiko Higashiguchi, Hiroki Tani, Kazuma Horiguchi, Masaki Furukawa, Masahiro Ito, Yutaro Kato, Tsunekazu Hanai, Akihiko Horiguchi

**Affiliations:** Department of Gastroenterological Surgery, Fujita Health University School of Medicine Bantane Hospital, 3-6-10 Otobashi Nakagawa Ward, Nagoya 454-8509, Aichi Prefecture, Japan; yoshiki.kunimura@fujita-hu.ac.jp (Y.K.); sarakawa@fujita-hu.ac.jp (S.A.); masasimu@fujita-hu.ac.jp (M.S.); takahiro.tashiro@fujita-hu.ac.jp (T.T.); daisuke.koike@fujita-hu.ac.jp (D.K.); hnagata@fujita-hu.ac.jp (H.N.); yuccakon@fujita-hu.ac.jp (Y.K.); h.yasuoka@aichi-cc.jp (H.Y.); thigashi@fujita-hu.ac.jp (T.H.); hiroyuki.tani@fujita-hu.ac.jp (H.T.); 81023034@fujita-hu.ac.jp (K.H.); masaki.furukawa@fujita-hu.ac.jp (M.F.); masito911@yahoo.co.jp (M.I.); y-kato@fujita-hu.ac.jp (Y.K.); kouichi@fujita-hu.ac.jp (T.H.); akihori@fujita-hu.ac.jp (A.H.)

**Keywords:** serum albumin levels, duodenum-preserving pancreatic head resection, remnant pancreatic volume

## Abstract

This retrospective study investigates the effects of duodenum-preserving pancreatic head resection (DPPHR) versus pancreaticoduodenectomy (PD) on postoperative pancreatic volume and nutritional parameters in patients with low-grade pancreatic malignancies. A total of 41 patients were analyzed, including 23 who underwent DPPHR and 18 who received PD. Postoperative pancreatic volume was measured using computed tomography on postoperative day 7 and at 1 year. Serum zinc and albumin levels, along with prognostic nutritional index (PNI), were also evaluated. The DPPHR group showed significantly better maintenance of remnant pancreatic volume at both time points compared to the PD group. Additionally, serum zinc and albumin levels were significantly higher in the DPPHR group at 1 year postoperatively. No significant difference in the PNI was observed. These findings suggest that DPPHR is advantageous in maintaining pancreatic tissue and supporting nutritional status after surgery, possibly due to preservation of the duodenum, which plays a key role in zinc absorption and digestive hormone regulation. This study emphasizes the functional and nutritional benefits of DPPHR compared to PD. Future prospective studies with larger cohorts and detailed functional assessments are necessary to validate these findings and explore their implications for long-term patient outcomes and quality of life.

## 1. Introduction

Duodenum-preserving pancreatic head resection (DPPHR) was first described by Beger et al. in 1980 as a surgical treatment for severe chronic pancreatitis [1]. However, at that time, surgical indications were limited to benign conditions, such as chronic pancreatitis, and neoplastic lesions were not considered eligible for this procedure. Subsequently, in 1993, Takada et al. [2] reported a modified DPPHR technique involving the complete resection of the pancreatic head. Unlike Beger’s first approach, which preserved the pancreatic head rim margin, this technique involves total excision of the pancreatic head and has expanded indications to include low-grade malignant tumors, such as intraductal papillary mucinous neoplasms (IPMNs) and other precancerous lesions. This method avoids the Kocher maneuver and preserves the gastroduodenal artery (GDA) and posterior superior pancreaticoduodenal artery (PSPDA), thereby maintaining blood flow to the duodenum. Following Takada’s procedure, we actively performed DPPHR for low-grade malignant tumors of the pancreatic head at our institution. Regarding nutritional aspects after this procedure, it has been reported that DPPHR results in better digestive and absorptive function and is less likely to cause nutritional deficiencies compared to pancreaticoduodenectomy (PD) [3]. Recently, several reports have suggested that zinc malabsorption after PD significantly contributes to postoperative nutritional deficiencies [4,5,6]. This is attributed to the fact that zinc is primarily absorbed in the third part of the duodenum and the proximal jejunum [7,8], and PD results in decreased zinc absorption due to duodenal resection. In contrast, DPPHR preserves the duodenum, suggesting that postoperative serum zinc levels may be maintained; however, no study has specifically investigated this aspect.

Kato et al. used an 80% pancreatectomy model in dogs to compare pancreatic regeneration between the groups with and without postoperative zinc supplementation [9]. Their findings indicated that pancreatic regeneration was significantly enhanced in the zinc-supplemented group 1 week postoperatively. However, no studies have compared postoperative pancreatic volumes between DPPHR and PD.

In this study, we aimed to compare the remnant pancreatic volume at postoperative 1 week and 1 year and the nutritional parameters between DPPHR and PD while also examining the potential impact of serum zinc levels, which may influence these outcomes.

## 2. Materials and Methods

This study included 41 patients who underwent subtotal stomach-preserving pancreaticoduodenectomy (SSPPD) or DPPHR for low-grade malignancies in our department between 2010 and 2021. The inclusion criteria were as follows: preoperative, postoperative one week (days 4–10, mean 7 days), and 12-month postoperative residual pancreatic volume measurable on computed tomography (CT) and the pancreatic transection line at the left border of the portal vein based on intraoperative findings. The patients were categorized into the DPPHR (*n* = 23) and SSPPD (*n* = 18) groups. Comparisons were made between the two groups regarding preoperative, early postoperative (days 4–10), and 12-month postoperative parameters. Additionally, nutritional indices at 12 months postoperatively and serum zinc levels, which have been reported to be frequently deficient after PD, were compared. Serum zinc levels were analyzed in patients for whom samples were available at 6 months postoperatively (DPPHR, *n* = 11; SSPPD, *n* = 7). No patients received zinc supplementation during the follow-up period. Therefore, the serum zinc levels evaluated in this study were not affected by exogenous zinc preparations.

The key technical aspect of DPPHR in our department involves complete resection of the pancreatic head without performing the Kocher maneuver. In this procedure, the anterior superior pancreaticoduodenal artery (ASPDA)–anterior inferior pancreaticoduodenal artery (AIPDA) arcade and the posterior superior pancreaticoduodenal artery (PSPDA)–posterior inferior pancreaticoduodenal artery (PIPDA) arcade are preserved. PSPDA preservation is particularly crucial for maintaining biliary blood flow. The PSPDA typically courses from the right side of the bile duct to the dorsal aspect of the bile duct and then traverses the posterior pancreas. The PIPDA, which frequently arises directly from the superior mesenteric artery (SMA), runs through the pancreatic uncinate fascia. The PIPDA was retained by avoiding Kocher’s maneuver and preserving the fascia. Digestive reconstruction was performed as follows. In both DPPHR and SSPPD, pancreaticojejunostomy was performed using the elevated jejunum, which was brought up 40 cm distal to the ligament of Treitz, via a retrocolic route. In the DPPHR group, the jejunojejunostomy was performed using the Roux-en-Y method with an end-to-side anastomosis. In the SSPPD group, reconstruction followed Child’s modified method (type II reconstruction) with an additional Braun anastomosis.

Pancreatic volume was calculated based on the method described by Sato et al. [10]. A 320-row CT scanner (Canon Aquilion ONE) was used to acquire serial contrast-enhanced axial images. The preoperative, predictive remnant pancreas, and remnant pancreas on postoperative 1 week and postoperative 1 year were traced on each slice, as shown in Figure 1A–D, and the total pancreatic tissue area was computed (Figure 1A). Major blood vessels such as the splenic vein and dilated pancreatic ducts (>3 mm) were excluded from the measurement. The predicted residual pancreatic volume was estimated retrospectively from operative records. The transection site was identified, and the corresponding preoperative CT images were analyzed to similarly calculate the pancreatic tissue area (Figure 1B).

Regarding the statistical analysis, continuous variables were compared between the two groups using Student’s *t*-test, whereas categorical variables were analyzed using the chi-square test. For the comparison of serum zinc levels, the Mann–Whitney U test was applied due to the small sample size and potential non-normal distribution. Statistical significance was set at *p* < 0.05.

## 3. Results

The baseline characteristics of the 41 patients were as follows: mean age, 64 years (range, 19–86 years); male-to-female ratio, 25:16. Primary diseases included intraductal papillary mucinous neoplasm (IPMN) in thirty-one cases, pancreatic neuroendocrine neoplasm (PNEN) in six cases, solid pseudopapillary neoplasm (SPN) in three cases, and serous cystic adenoma (SCA) in one case. A comparison of background factors between the two groups is presented in Table 1.

Regarding preoperative factors, the mean age of the DPPHR group was significantly lower than that of the pancreaticoduodenectomy (PD) group (54.4 ± 15.8 vs. 70.0 ± 9.2 years, *p* = 0.001). No significant differences were observed between the two groups in terms of intraoperative factors, including operative time and blood loss. Postoperative complications classified as Clavien–Dindo grade III or higher occurred in 34% of the DPPHR group and 33% of the PD group (*p* = 0.923). The incidence of postoperative pancreatic fistulae was 26.1% in the DPPHR group and 27.8% in the PD group (*p* = 0.903), with no significant difference between the surgical procedures.

When comparing preoperative pancreatic volume and predicted remnant pancreatic volume, no significant differences were observed between the DPPHR and PD groups (51.3 vs. 57.7 cc, *p* = 0.376; 33.5 vs. 37.3 cc, *p* = 0.327). However, when evaluating the change in pancreatic volume at 1 week and 1 year postoperatively, the DPPHR group demonstrated significantly better pancreatic volume preservation than the PD group (31.7 vs. 27.8 cc at 1 week, *p* = 0.045; 24.5 vs. 20.2 cc at 1 year, *p* = 0.041) (Figure 2).

Comparison of the volume maintenance ratio, defined as the ratio of the remnant pancreatic volume at 1 week and 1 year postoperatively to the predicted remnant pancreatic volume, revealed significantly better pancreatic volume preservation in the DPPHR group than in the PD group (96.7% vs. 78.0% at 1 week, *p* = 0.045; 79.1% vs. 58.6% at 1 year, *p* = 0.041) (Figure 3).

Focusing on postoperative nutrition as shown in Figure 4, the serum albumin levels at one year were significantly higher in the DPPHR group compared to the PD group (4.27 vs. 4.01 g/dL, *p* = 0.001). However, there was no significant difference in the prognostic nutritional index (PNI) between the two groups (49.4 vs. 48.1, *p* = 0.284).

The postoperative serum zinc levels were significantly higher in the DPPHR group than in the PD group (80.3 vs. 65.8 μg/dL, *p* = 0.002), suggesting better preservation of zinc levels in the DPPHR group (Figure 5).

## 4. Discussion

Our study revealed the following three novel findings: (1) In the DPPHR group, the remnant pancreatic volume on postoperative day 7 and at 1 year was significantly larger than that in the PD group. Furthermore, the volume maintenance ratio calculated from the predicted remnant pancreas was significantly higher in the DPPHR group. (2) Serum albumin levels 1 year postoperatively were significantly higher in the DPPHR group. However, there were no significant differences in the prognostic nutritional index (PNI). (3) Serum zinc levels, which are primarily reabsorbed from the horizontal portion of the duodenum to the proximal jejunum, were significantly higher in the DPPHR group.

Pancreatic volume, particularly remnant pancreatic volume, is strongly correlated with both exocrine and endocrine functions [10,11]. This reduction was a reliable indicator of a functional decline. For instance, in chronic pancreatitis, pancreatic atrophy leads to a decreased secretion of digestive enzymes and fatty diarrhea, resulting in malnutrition. The association between pancreatic atrophy and exocrine insufficiency after PD has been well-documented. Okano et al. reported that in 227 patients undergoing PD or distal pancreatectomy, those with a remnant pancreatic volume of <24.1 mL had a significantly higher risk of developing pancreatic exocrine insufficiency (PEI) [11]. Similarly, Tsunematsu et al. [12] found a relationship between the onset of nonalcoholic fatty liver disease (NAFLD) after PD and estimated functional remnant pancreatic volume. They reported that a low “pancreatic exocrine index”, combining remnant pancreatic volume and pancreatic juice amylase activity, was associated with a higher risk of NAFLD. Our department also reported that the incidence of NAFLD was significantly lower after DPPHR than after PD [13]. However, to the best of our knowledge, this is the first study to directly compare the change in remnant pancreatic volume between DPPHR and PD patients from immediately after surgery to postoperative 1 year. In this study, a significant difference in remnant pancreatic volume was observed on postoperative day 7. The exact cause of this remains unclear; however, unlike in PD, patients with DPPHR do not undergo hepaticojejunostomy. Therefore, in Roux-en-Y reconstruction, bile and pancreatic juice merge at the jejunojejunostomy site, reducing the activation of pancreatic enzymes at the pancreaticojejunostomy site and potentially minimizing inflammation and edema at the anastomosis site. While this remains a hypothesis, the favorable volume preservation at 1 year in the DPPHR group may support this theory.

In DPPHR, the following three factors are thought to contribute to the long-term postoperative preservation of remnant pancreatic volume compared to PD. First, preservation of the duodenum, where zinc is absorbed, allows for the postoperative maintenance of zinc absorption efficiency. According to Kato et al., animals fed a high-zinc diet after 80% pancreatectomy maintained pancreatic zinc levels, which significantly increased ODC activity, DNA synthesis, and PCNA positivity, thereby promoting pancreatic regeneration [9]. Therefore, duodenal preservation by DPPHRs likely supports cell proliferation and volume maintenance in the remnant pancreas. Second, gastrointestinal hormones involved in pancreatic secretion (e.g., secretin and cholecystokinin) are preserved postoperatively in DPPHR, promoting early recovery of exocrine function and preventing atrophy. While this study used CT-based volumetric analysis, it is recognized that anatomical volume does not always reflect exocrine functional capacity. In our department, we utilize the 13C-trioctanoin breath test as a non-invasive assessment tool for pancreatic exocrine function [14]. We have previously demonstrated through this method that patients undergoing DPPHR maintained significantly better exocrine function compared to those who underwent PD, as evidenced by better intestinal fat absorption levels [15]. This functional advantage reinforces the clinical value of duodenum preservation.

Furthermore, the duodenal mucosa contains important hormone-secreting cells and receptors—S cells and I cells—which produce secretin and cholecystokinin (CCK) in response to gastric acid and luminal nutrients, respectively. These hormones stimulate bicarbonate and enzyme secretion from the pancreas [16]. Thus, preserving the duodenum may help maintain this physiological feedback mechanism and contribute to better long-term pancreatic function [17].

Although secretin-stimulated MRCP (S-MRCP) is a promising imaging modality for evaluating pancreatic exocrine function dynamically [18,19,20,21], this approach is currently not feasible in Japan, as secretin is not commercially available or approved for clinical use. Therefore, as above described, the use of the 13C breath test provides a practical and validated alternative in the clinical setting [22,23].

Taken together, these findings support the hypothesis that duodenal and bile duct preservation contribute not only to anatomical volume maintenance but also to physiological function of the remnant pancreas. Future prospective studies should aim to integrate both imaging-based volumetry and functional assessments to further validate the long-term outcomes of DPPHR.

Third, the physiological digestion and absorption of nutrients are preserved, enhancing the utilization efficiency of trace elements such as zinc and ensuring sufficient nutritional supply for pancreatic tissue reconstruction. These combined effects likely contributed to the maintenance of pancreatic volume after DPPHR.

Previous reports have shown that DPPHR is associated with the better preservation of exocrine and endocrine functions and favorable nutritional status. Miyakawa et al. reported that fat absorption, assessed using the 13C breath test, was significantly better with DPPHR than with standard PD [24]. Horiguchi et al. also found superior fat absorption in DPPHR compared with PPPD [15]. Recent advances in minimally invasive surgery (MIS) further support these findings, with similar results reported for laparoscopic DPPHR and PD [25]. Although our findings reproduce these trends, the small sample size warrants the further accumulation of more cases. Moreover, the lack of a significant difference in the PNI may be attributed to lower lymphocyte counts in the DPPHR group, which requires further investigation.

The pancreas plays a crucial role in zinc metabolism, with high zinc concentrations in the pancreatic juice [26]. After PD, the resection of the proximal jejunum, which plays a critical role in zinc absorption [27], may lead to reduced serum zinc levels and impaired exocrine function. In contrast, DPPHR preserves the duodenum and proximal jejunum, thus maintaining zinc absorption. Zinc deficiency can exacerbate malnutrition. It causes taste disorders, reduces food intake, delays the turnover of intestinal epithelial cells, impairs absorption, suppresses immune function, and increases the risk of infection. This results in a vicious cycle of malnutrition [28,29,30,31]. Thus, the preservation of zinc absorption via DPPHR is likely to contribute to the maintenance of nutritional status. Although comorbidities such as diabetes mellitus and chronic pancreatitis are known to affect serum zinc levels, none of the patients in our zinc-evaluated subgroup had these conditions during the study period, thereby minimizing confounding influences, though further large-scale studies are needed to exclude these confounding factors.

This study had several limitations. First, it was a retrospective study with a relatively small sample size, which may have limited the generalizability of the findings. Second, serum zinc levels were not measured in all patients, and the timing of the zinc measurement was not standardized, introducing potential variability in the data. Third, due to the retrospective nature of this study, the timing of postoperative CT scans for the remnant pancreatic volume assessment was not uniformly determined, potentially introducing a measurement bias. These limitations should be considered when interpreting the results. Prospective studies with larger cohorts and standardized protocols are necessary to validate these findings.

## 5. Conclusions

Preservation of the duodenum in DPPHR may help maintain the remnant pancreatic volume and serum zinc levels postoperatively, potentially preventing nutritional decline and contributing to improved patient quality of life.

## Figures and Tables

**Figure 1 cancers-17-02217-f001:**
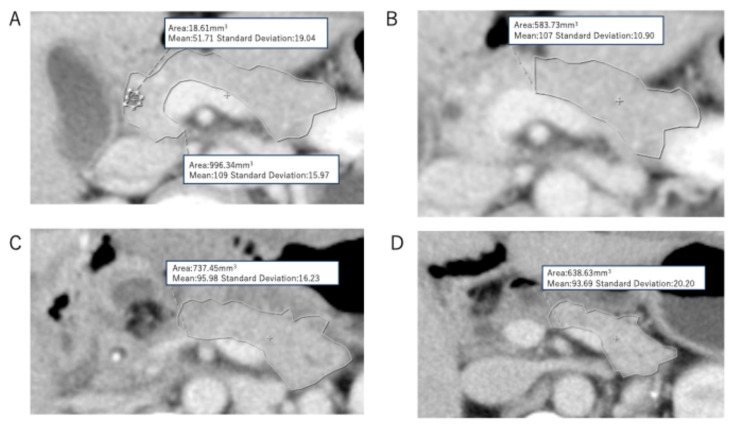
CT volumetry of pancreas. (**A**) Preoperative status; (**B**) predictive remnant pancreatic volume; (**C**) remnant pancreatic volume at the time of postoperative one week; (**D**) remnant pancreatic volume at the time of postoperative one year.

**Figure 2 cancers-17-02217-f002:**
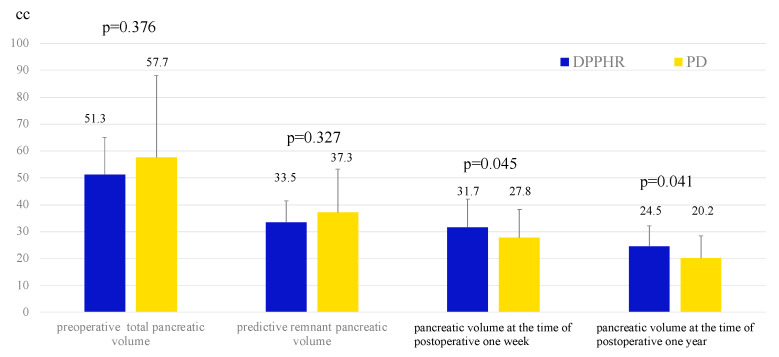
Comparison of pancreatic volume between the DPPHR and PD groups at the time of preoperative, predictive, postoperative one week, and postoperative one year, respectively (Student’s *t*-test).

**Figure 3 cancers-17-02217-f003:**
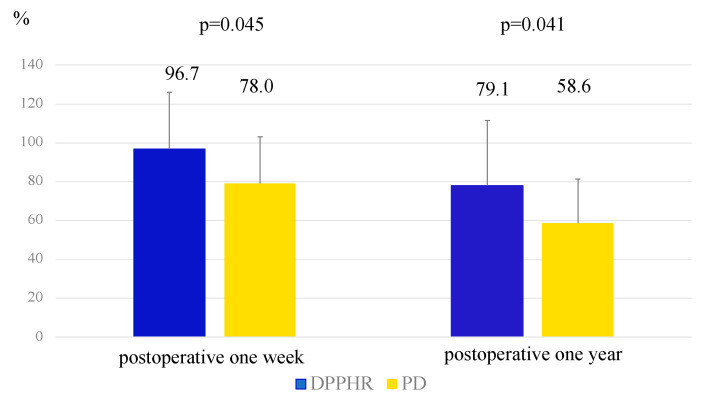
Comparison of volume maintenance ratio between the DPPHR and PD groups at the time of postoperative one week and year, respectively (Student’s *t*-test).

**Figure 4 cancers-17-02217-f004:**
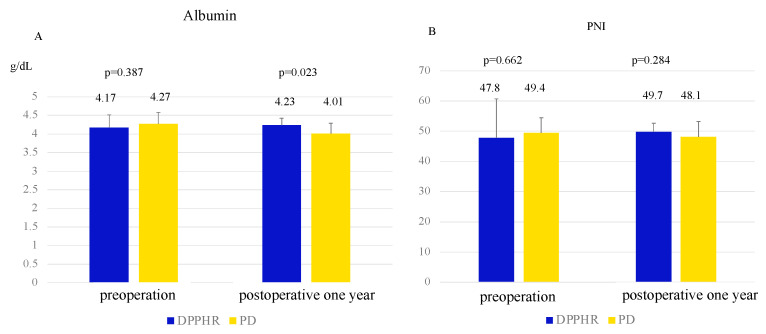
Comparison of the nutritional marker between the DPPHR and PD groups at the time of POY1. (**A**). Serum albumin level. (**B**). Prognostic nutrition index (PNI) (Student’s *t*-test).

**Figure 5 cancers-17-02217-f005:**
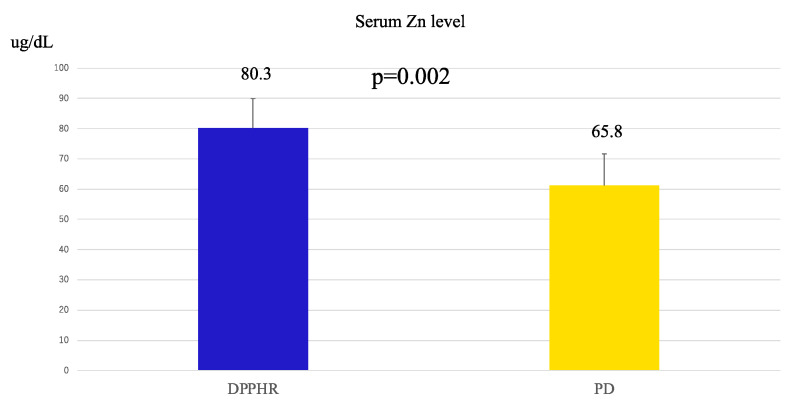
Comparison of serum Zn level between the DPPHR and PD groups at the time of postoperative one week (Mann–Whitney U test).

**Table 1 cancers-17-02217-t001:** Comparison of patient background between the DPPHR and PD groups.

	DPPHR (*n* = 23)	PD (*n* = 18)	*p*-Value
**Preoperative variables**			
Age (years)	54.4 ± 15.8	70.0 ± 9.2	0.001
Gender (male/female)	11/12	14/4	0.051
Body mass index	20.3 ± 3.1	22.4 ± 3.0	0.110
Primary disease			
intraductal papillary mucinous adenoma	12	10	
intraductal papillary mucinous carcinoma (non-invasive)	1	8	
neuroendocrine neoplasm (WHO classification grade1)	6	0	
solid pseudopapillary neoplasm (borderline malignancy)	3	0	
serous cystic adenoma	1	0	
Albumin (g/dL)	4.17 ± 0.35	4.27 ± 0.31	0.380
Total lymphocyte count (/mm^3^)	1775 ± 678	1554 ± 609	0.360
Hemoglobin (g/dL)	13.8 ± 1.0	13.6 ± 1.4	0.729
Prognostic nutrition index	47.8 ± 12.9	49.4 ± 5.1	0.377
Aspartate aminotransferase (AST) (IU/L)	21.5 ± 6.1	21.2 ± 14.9	0.872
Alanine transaminase (ALT) (IU/L)	19.2 ± 10.0	18.4 ± 6.1	0.797
Platelet count (10^3^/μL)	20.2 ± 5.3	27.9 ± 14.4	0.076
**Intraoperative variables**			
Operative time (min)	477 ± 87.3	438 ± 96	0.193
Blood loss (g)	379 ± 452	284 ± 267	0.448
**Postoperative variables**			
Clinically relevant pancreatic fistula (%)	26.1% (6/23)	27.8% (5/18)	0.903
Complication with Clavien-Dindo grade ≥3 (%)	34.8% (8/23)	33.3% (6/18)	0.923

DPPHR: duodenum-preserving pancreatic head resection, PD: pacreaticoduodenectomy Data are expressed as mean ± SD. Comparisons were made using Student’s *t*-test or chi-square test.

## Data Availability

All the data generated or analyzed during this study are included within the article.

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
