# Peer review of "Clinical Influence of Bile Duct and Duodenum Preservation on Zinc Absorption and Remnant Pancreatic Volume in Duodenum-Preserving Pancreatic Head Resection for Low-Grade Malignant Pancreatic Tumors"

_cancers, 2025, doi:10.3390/cancers17132217_

Round 1

Reviewer 1 Report

Comments and Suggestions for Authors

The authors are right to emphasize the importance of preserving the duodenum after resection of the head of the pancreas, but I must also emphasize the role of the hormone receptors present on the duodenal mucosa that are necessary to maintain the active function of the remnant pancreas. Furthermore, they should know that the radiological study of the pancreatic volume must be correlated with its function and therefore it is better to study the function with secretin stimulation and dynamic evaluation with MRI, than with only Ct scan. 

Author Response

Comment 1:
The authors are right to emphasize the importance of preserving the duodenum after resection of the head of the pancreas, but I must also emphasize the role of the hormone receptors present on the duodenal mucosa that are necessary to maintain the active function of the remnant pancreas. Furthermore, they should know that the radiological study of the pancreatic volume must be correlated with its function and therefore it is better to study the function with secretin stimulation and dynamic evaluation with MRI, than with only Ct scan.

Response:
We appreciate the reviewer’s comments regarding the hormonal and functional aspects of duodenal preservation. In response, we have revised the Discussion section to include a detailed explanation of the importance of hormone receptors, such as those for secretin and CCK, which are distributed on the duodenal mucosa and contribute to the regulation of remnant pancreatic function. We also acknowledge the limitations of CT-based volume assessment and have now discussed the potential benefits of secretin-enhanced dynamic MRI as a more functional and physiologically relevant modality. Unfortunately, this imaging modality is currently not feasible in Japan due to the unavailability of secretin for clinical use. Therefore, we have added the following statement to the Discussion section (lines227-249) :

While this study used CT-based volumetric analysis, it is recognized that anatomical volume does not always reflect exocrine functional capacity. In our department, we utilize the 13C-trioctanoin breath test as a non-invasive assessment tool for pancreatic exocrine function[15]. We have previously demonstrated through this method that patients undergoing DPPHR maintained significantly better exocrine function compared to those who underwent PD, as evidenced by better intestinal fat absorption levels[16]. This functional advantage reinforces the clinical value of duodenum preservation.

Furthermore, the duodenal mucosa contains important hormone-secreting cells and receptors—S cells and I cells—which produce secretin and cholecystokinin (CCK) in response to gastric acid and luminal nutrients, respectively. These hormones stimulate bicarbonate and enzyme secretion from the pancreas[17]. Thus, preserving the duodenum may help maintain this physiological feedback mechanism and contribute to better long-term pancreatic function[18].

Although secretin-stimulated MRCP (S-MRCP) is a promising imaging modality for evaluating pancreatic exocrine function dynamically[19-21], this approach is currently not feasible in Japan, as secretin is not commercially available or approved for clinical use. Therefore, the use of the 13C-trioctanoin breath test provides a practical and validated alternative in our clinical setting, as above described.

Taken together, these findings support the hypothesis that duodenal and bile duct preservation contributes not only to anatomical volume maintenance but also to physiological function of the remnant pancreas. Future prospective studies should aim to integrate both imaging-based volumetry and functional assessments to further validate the long-term outcomes of DPPHR.

Reviewer 2 Report

Comments and Suggestions for Authors

The present study is a retrospective study regarding to DPPHR and PD. The authors compared both surgery types by serum zinc level, which may be linked to the pancreas reserve volume. There are several points need to be clarified.

1.The characteristics of the patients groups are not enough, despite of a “low grade pancreas malignancy” is defined in the present study. The stage of cancer or other etiologies of patients groups would be added in table 1.

2.Co-morbidities status, such as diabetes or chronic pancreatitis, would also be linked to the serum zinc level. Thus these factors would be also mentioned.

3.Due to limited case numbers, the student’s t test is not adequate, particularly in figure 5. I suggested a Mann-Whitney U test for this.

  1. the zinc formula is also a confounder, how could the authors to manage in the study.

5.pancreas enzyme also affects the serum zinc level, could the authors excluded this confounding factor.

6.Is serum Zinc level correlated to the prognosis or survival or QOL ?

  1. If key confounding factors cannot be adequately controlled, the authors are advised to focus on evaluating pancreatic volume and surgical approach rather than serum zinc levels as the primary outcome.

8.the abbreviations of the figure would be afflicted at the bottom of the table, such as POD, POY. Also the statistic methods would be mentioned.

Author Response

Comment 1:

1.The characteristics of the patients groups are not enough, despite of a “low grade pancreas malignancy” is defined in the present study. The stage of cancer or other etiologies of patients groups would be added in table 1.

Response:
Among the 31 patients diagnosed with intraductal papillary mucinous neoplasm (IPMN), 22 were classified as intraductal papillary mucinous adenoma (IPMA) and 9 as intraductal papillary mucinous carcinoma (IPMC), all of which were intraepithelial lesions without invasion. All cases of neuroendocrine tumor (NET) were grade 1 according to the WHO classification. Serous cystic adenoma (SCA) cases were confirmed as benign. In addition, 3 patient had a solid pseudopapillary neoplasm (SPN), which is considered a borderline malignant tumor.

Importantly, all tumors included in this study were stage I or lower, and no cases exhibited invasive cancer. This information has been added to Table 1 and clarified in the revised manuscript.

Comment 2:

Co-morbidities status, such as diabetes or chronic pancreatitis, would also be linked to the serum zinc level. Thus these factors should be mentioned.

Response:

We appreciate the reviewer’s concern regarding potential confounding comorbidities that could affect serum zinc levels. Among the 18 patients for whom serum zinc levels were evaluated, none had a history of chronic pancreatitis. Additionally, none of the patients required insulin therapy after surgery during the follow-up period. Therefore, we believe that the impact of these co-morbidities on the zinc data in this study is minimal. This clarification has been added to the revised manuscript ( line272-276).

Although comorbidities such as diabetes mellitus and chronic pancreatitis are known to affect serum zinc levels, none of the patients in our zinc-evaluated subgroup had these conditions during the study period, thereby minimizing confounding influences, whereas further large-scale study should be needed to exclude these confounding factors.  

Comment 3:

Due to limited case numbers, the student’s t test is not adequate, particularly in figure 5. I suggested a Mann-Whitney U test for this.

We totally agree with thiscomment and reanalyzed the data using the Mann-Whitney U test in place of Student’s t test for comparisons in Figure 5. The methods and figure legends have been updated accordingly (Line 126-128).

For the comparison of serum zinc levels shown in Figure 5, the Mann–Whitney U test was applied due to the small sample size and potential non-normal distribution.

Comment 4:

The zinc formula is also a confounder, how could the authors to manage in the study.

Response:

We thank the reviewer for pointing this out. None of the patients whose zinc level was evaluated received zinc supplementation during the observation period. Therefore, the influence of different zinc formulas or preparations on serum zinc levels can be excluded. This information has been added to the Methods section (Line95-97).

No patients received zinc supplementation during the follow-up period. Therefore, the serum zinc levels evaluated in this study were not affected by exogenous zinc preparations.

Comment 5:

pancreas enzyme also affects the serum zinc level, could the authors excluded this confounding factor.

Response:

We appreciate the reviewer’s important observation. In our institution, all patients received standardized pancreatic enzyme supplementation during the perioperative and follow-up periods according to our clinical pathway. Therefore, we believe that the influence of pancreatic enzyme insufficiency on serum zinc levels was minimized and consistent across both groups.

Comment 6:

Is serum Zinc level correlated to the prognosis or survival or QOL?

Response:

We thank the reviewer for comment. Among the 18 patients in whom serum zinc levels were measured, all survived during the follow-up period. Therefore, we were unable to analyze correlations between zinc levels and prognosis or overall survival. Although this study did not directly assess quality of life (QOL), it is known that zinc deficiency is associated with symptoms such as taste disturbance and poor wound healing, which can negatively impact QOL. Thus, maintaining serum zinc levels after surgery may contribute to improved postoperative QOL.

Comment 7:

If key confounding factors cannot be adequately controlled, the authors are advised to focus on evaluating pancreatic volume and surgical approach rather than serum zinc levels as the primary outcome.

Response:

We understand the reviewer’s concern regarding potential confounding factors related to serum zinc levels. However, as noted above, key confounding variables such as diabetes mellitus, chronic pancreatitis, pancreatic enzyme supplementation, and zinc intake were either absent or consistently managed across all patients included in the zinc evaluation subgroup. Therefore, we believe that serum zinc levels can be appropriately interpreted in this context. Given the physiological importance of zinc absorption in the duodenum and its relationship to pancreatic function and nutrition, we consider serum zinc level to be a meaningful parameter and have chosen to retain it as one of the primary outcomes in this study.

Comment 8:

The abbreviations of the figure would be afflicted at the bottom of the table, such as POD, POY. Also the statistic methods would be mentioned.

Response:

We have updated all figures and tables to include full descriptions of abbreviations and statistical methods used, in accordance with journal guidelines